# Single-cell multiomic human brain atlas reveals regulatory drivers of cortical regionality

Carter R. Palmer[1,2,12], Jinghui Song [3,12], Bing Yang[4,5,12], Chien-Ju Chen[3,12], Dinh Diep[3], Kimberly Conklin[3], Nongluk Plongthongkum[3], Hannah S. Indralingam[4,5], Christine S. Liu [1,2], Joshua Kurtz [1], Qiwen Hu[6], Linnea Ransom[1,2], Anis Shahnaee[1], Annie Hiniker[7], Rebecca D. Hodge [8], C. Dirk Keene [9], Ed Lein [8], Peter Kharchenko[6,10], Nathan R. Zemke[4,5] ✉, Jerold Chun [1] ✉, Bing Ren[4,5,11] ✉ & Kun Zhang [3,10] ✉

Distinct regional functionality of the human cortex is orchestrated by diverse cellular and molecular processes, yet the underlying regulatory mechanisms remain poorly understood. We performed multiomic single-cell and spatial characterization of nine regions of the human cortex to define the gene regulatory networks and transcription factors that govern cell-type and region specificity. With the combined data of over three million cells, two striking patterns of cortical neuron specialization were uncovered: a rostral-caudal spatial pattern of calcium regulatory machinery, and subunit switching of multiple signaling receptor families across the transmodal-sensory axis. Gene regulatory network analysis revealed putative transcriptional regulators of cortical neuron specialization with cell-type- and region-specific gene regulation patterns. While regionalization was observed in gene expression, chromatin accessibility, and spatial distributions, these modalities exhibited distinct cortical patterns. Our findings illuminate critical neuronal pathways that vary throughout the cortex and the gene regulatory networks that establish cortical regionalization in the human brain.

The human cortex is intricately organized into distinct regions that support specialized brain functions. Underpinning these region-specific functional changes are variations of the cellular and molecular composition of the brain. Advanced high-throughput sequencing technologies have resulted in an unparalleled opportunity to study the composition of the human brain on a single-cell level. Single-nucleus RNA-sequencing studies have precisely identified the types of cells within the human brain[1–4] and have been augmented to compare how these cell types vary across mammalian evolution[2,5,6]. Moreover, the epigenetic profiles of these cells have been carefully detailed across

[1]Center for Neurological Diseases, Sanford Burnham Prebys Medical Discovery Institute, La Jolla, CA, USA. [2]Biomedical Sciences Program, School of Medicine, University of California, San Diego, La Jolla, CA, USA. [3]Department of Bioengineering, University of California, San Diego, La Jolla, CA, USA. [4]Department of Cellular and Molecular Medicine, University of California, San Diego, La Jolla, CA, USA. [5]Center for Epigenomics, University of California, San Diego, School of Medicine, La Jolla, CA, USA. [6]Department of Biomedical Informatics, Harvard Medical School, Boston, MA, USA. [7]Department of Pathology, University of Southern California, Los Angeles, CA, USA. [8]Allen Institute for Brain Science, Seattle, WA, USA. [9]Department of Laboratory Medicine and Pathology, University of Washington, Seattle, WA, USA. [10]San Diego Institute of Science, Altos Labs, San Diego, CA, USA. [11]Present address: New York Genome Center, New York NY, USA. [12]These authors contributed equally: Carter R. Palmer, Jinghui Song, Bing Yang, Chien-Ju Chen. ✉e-mail: nzemke@health.ucsd.edu; jchun@SBPdiscovery.org; bren@nygenome.org; kzhang@altoslabs.com

mouse[7,8] and human[9,10] brains. Spatial transcriptomics has enabled further characterization of cell localizations across the brain[3,5,11,12].

These single-cell transcriptomic and epigenetic studies have characterized changes to cellular expression and epigenetic state across the cortical regions, yet identification of the molecular mechanisms that drive functional specialization as well as the gene regulatory networks (GRNs) responsible for these differences remain underexplored. While cortical cellular variability has been shown to change across distinct axes[3,13,14], the regulatory mechanisms behind these changes are unknown. Delineating the regulatory frameworks that shape cell-type-specific organizational and functional changes across the cortex is essential for elucidating fundamental principles of cortical dynamics and disease susceptibility.

To elucidate the molecular architecture governing regional specialization across the cortex, we generated combined single-nucleus profiles of both the transcriptome and the epigenome for over three million cells and characterized spatial transcriptomics across nine regions of the human cortex. We identified distinct expression, chromatin accessibility, and spatial organization patterns across cortical regions, with intratelencephalic projecting neurons displaying the greatest regional specialization. Cellular and molecular composition varied across the rostral-caudal and transmodal-sensory axes of the human cortex and integrated analyses of transcriptomic and chromatin accessibility data enabled the identification of distinct transcription factors driving this variation.

Close examination of molecular changes revealed receptor subunit switching and changes in neuronal calcium dynamics as key features of cortical regionalization with distinct implications for neuronal signaling and plasticity as well as the pathophysiology of neurodevelopmental and neurodegenerative disorders. Transcription factors implicated in activating these pathways showed clear correlations in their expression and chromatin binding activities, matching that of their predicted target genes, implicating finely controlled transcriptional regulators responsible for the molecular variation that enables region specific functions. Our findings provide insights into the molecular adaptations that support regionalized local circuit demands in the human cortex.

## Results

### Single-cell multiomic analyses of the human cortex

To reconstruct GRNs responsible for cellular diversity across the human cortex, we profiled nine distinct brain regions via combined single-nucleus RNA-seq and ATAC-seq profiling as well as spatial transcriptomics. We profiled the anterior cingulate cortex (ACC), dorsolateral prefrontal cortex (DFC), frontal insular cortex (FI), motor cortex (M1C), somatosensory cortex (S1C), middle temporal gyrus (MTG), auditory cortex (A1C), angular gyrus (AnG), and visual cortex (V1C) regions that spatially and functionally span the breadth of the human neocortex (Fig. 1a and data S1). Cortical areas were carefully dissected from six donors postmortem using anatomical landmarks, and regional cytoarchitecture was confirmed via Nissl staining (Fig. 1b).

Single-nucleus datasets were generated using SNARE-seq2, a dual-omic profiling technique enabling the simultaneous profiling of accessible chromatin and nuclear RNA through combinatorial barcoding[15]. All cortical layers were profiled for each section to obtain high-resolution cellular datasets reflecting RNA and accessible chromatin across the cortex. Over three million nuclei were profiled, and after quality filtering (see methods) over 1.7 million high-quality nuclei were retained for downstream analysis (Supplementary Fig. 1a). Nuclei were clustered into 24 cell subclasses (Fig. 1c) with nomenclature (data S1) consistent with previous studies from the Brain Initiative Cell Census Network (BICCN)[3,16] via Seurat V5[17]. Excitatory neurons were classified as intratelencephalic (IT) projecting (L2/3 IT, L4 IT, L5 IT, L6 IT, L6 IT Car3) or deep layer (L5 ET, L5/6 NP, L6B, and L6 CT). Inhibitory neurons included medial ganglionic eminence (MGE)- derived

(Chandelier, PVALB, SST, and SST CHODL) and caudal ganglionic eminence (CGE)-derived (LAMP5, LAMP5 LHX6, VIP, SNCG, and PAX6) cells. Non-neurons (Astro, Micro/PVM, OPC, Oligo, Endo, and VLMC) were also classified. Quality control metrics, including UMIs and genes per cell showed high quality analysis and minimal variation across subclass, region, or donor (Supplementary Fig. 1b and c). Donor-specific differences were minimal for RNA-driven UMAP-based clustering (Supplementary Fig. 1d).

IT projecting neurons from the visual cortex were distinct from other populations via combined regional UMAPs from RNA data (Supplementary Figs. 1e and 2a). Subclass marker genes retained conserved patterns across cortical regions (Supplementary Fig. 3a and b and Supplementary Data 1), although the relative abundance of certain subclasses varied slightly by region (Fig. 1d) as expected[3]. As previously observed[3], the ratio of excitatory to inhibitory neurons increased across the rostral-caudal axis with a significant elevation in the visual cortex (Supplementary Fig. 3c) and distinct changes in specific subclass populations including L6 IT, L6B, SST CHODL, and PAX6 neurons (Supplementary Fig. 3d-f) were identified. Clustering identified 120 cell types (Fig. 1e) in accordance with previous efforts[3] and offered insight into V1C enriched glutamatergic cell types as well as the significant cell-type diversity contained within SST and VIP subclasses.

We used edgeR[18] to identify differentially expressed genes (DEGs) across cortical regions in distinct cell subclasses. Corroborating previous analyses[3], the most significant changes were observed in IT projecting and deep layer excitatory neurons followed by PVALB, and SST inhibitory neurons when subclasses were uniformly down sampled prior to analysis (Fig. 1f) or when all nuclei were analyzed (Supplementary Fig. 4a and Supplementary Data 2). This trend also held when the number of DEGs were normalized by the total number of genes sequenced in each subclass (Supplementary Fig. 4b). ACC and V1C contained the majority of region-specific DEGs, and several distinct expression modules were observed in rostral regions ACC, DFC, and FI as well as across motor and visual cortices (Fig. 1g). To account for the possibility that pseudobulk approaches can be underpowered we utilized MAST[19] to account for age, sex, PMI, and RIN effects on differentially expressed genes (Supplementary Data 3). As expected, MAST identified more regional DEGs, 21,234 as compared to 4,101 with edgeR, but 69.9% of the edgeR genes were also called as regional DEGs by MAST, confirming the robust nature of the analysis. To interrogate if specific gene programs were enriched in DEGs, we performed gene set enrichment analysis using clusterProfiler[20] on combined region-specific DEGs for each individual subclass for edgeR called DEGs. Various subclasses shared enriched gene sets associated with neuronal guidance and organization (Fig. 1h and Supplementary Fig. 4c and d). Previous efforts[21] have established regionality in gene expression in the developing human brain. To establish the extent to which regional gene expression is conserved from development to adulthood we determined the overlapping gene set from neuronal region-specific expression during development[21] against all neuron subclasses from this dataset. 5-15% of region-specific genes from this dataset were also identified as enriched in the same region of the developing human brain (Supplementary Fig. 4e and Supplementary Data 2), including the transcription factors *NFIA*, *NFIX*, *TCF4*, and *NR2F1*.

### Analyses of accessible chromatin across the human cortex

Our SNARE-seq2 data resulted in a three-fold increase in the number of neurons profiled for accessible chromatin in the human cortex as compared to previous efforts[10,22] (Fig. 2a). Accessible chromatin exhibited clear subclass-specific patterns near marker gene loci (Supplementary Fig. 5a). Quality metrics were similar across subclass, region, and donor (Supplementary Fig. 5b-d). Transcription start site (TSS) enrichment of accessibility was lower in neuronal populations

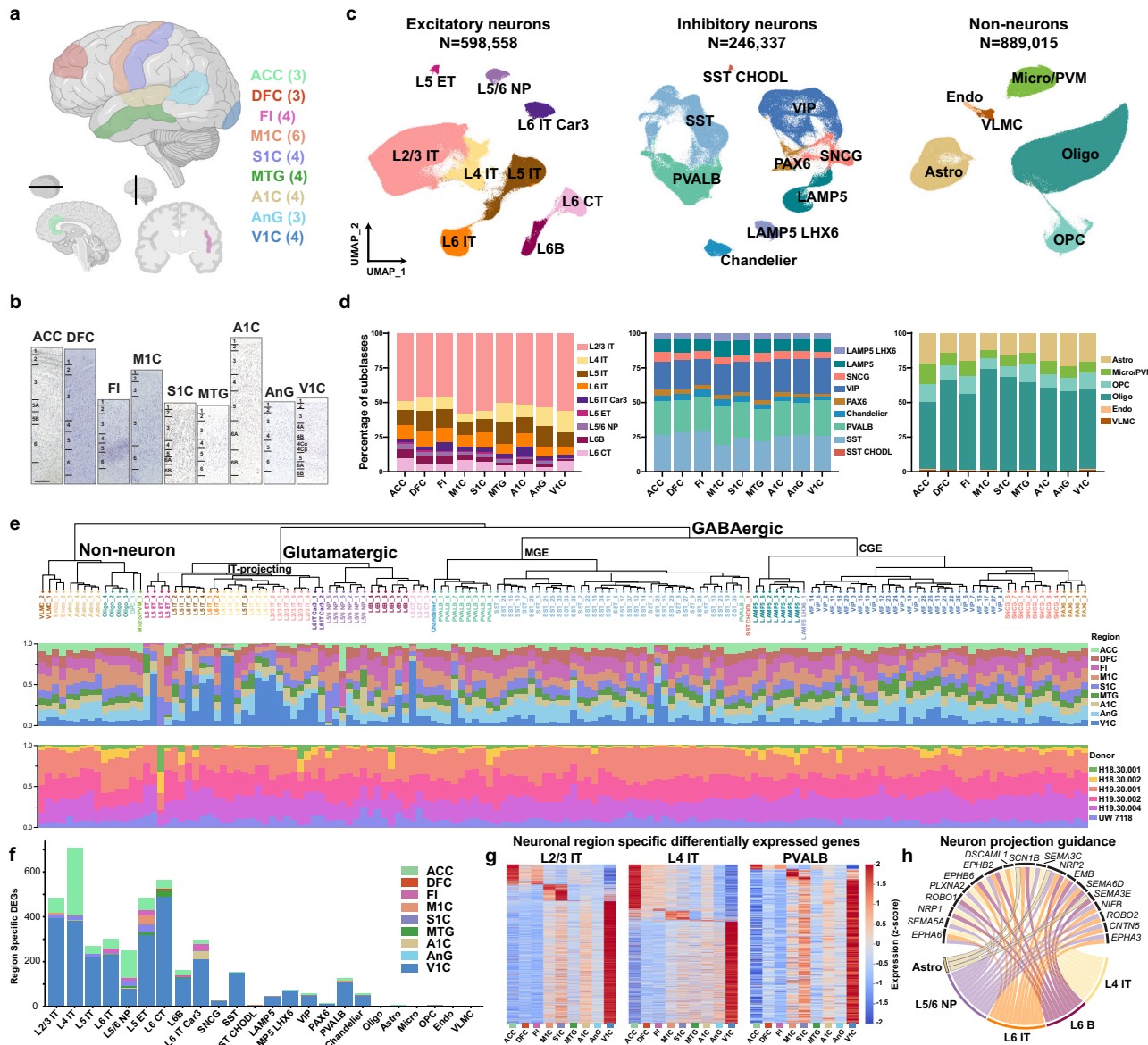

**Fig. 1 | Single-cell multiomic analyses of the human cortex. a** Schematic of 9 profiled cortical regions, created in BioRender. Costantino, I. (2026) https://BioRender.com/utnjof0. **b** Nissl-stained images from each cortical region with approximate layer boundaries. Scale bar is 500µm. Staining was repeated two to three times per region with similar results. **c** RNA-based UMAP embedding of each cell class based on single-nucleus RNA data. Colors correspond to subclass. Cell counts are profiled cells remaining after filtering. Detailed cell subclass information is provided in data S1. (**a**) was created with the use of Biorender.com. **d** Proportions of subclasses within each cell class organized by region. **e** Consensus taxonomy of cell types across 9 regions. Proportions of nuclei in each cell type dissected from each region (middle) and each donor (bottom) are shown. **f** Total counts of region-specific differentially expressed genes (DEGs) calculated with edgeR as genes differentially expressed in one region as compared to all others combined with uniform cell counts for a uniformly down sampled number of cells for each subclass. Plots are organized by cell type and divided by region. Missing subclasses had zero region-specific DEGs. **g** Heatmaps of region-specific differentially expressed genes for key neuronal subclasses. Z-score calculated independently for each subclass. **h** Cell subclasses and genes for which neuron projection guidance was an enriched term across the subclasses' region-specific DEGs. Only genes for which differential expression was observed in two or more cell subclasses are shown. Source data are provided as a Source Data file.

and elevated in non-neurons (Supplementary Fig. 5e), whereas total unique fragment counts were consistently elevated in neurons (Supplementary Fig. 5f) but both were consistent across region and donor.

To identify regulatory elements across the human cortex, candidate cis-regulatory elements (cCREs) were called by aggregating accessible chromatin signals for each subclass and utilizing MACS2[23] for peak calling. Both established and novel cCREs were identified (Fig. 2b), including subclass- (Fig. 2c), class-, and type-specific cCREs (Supplementary Fig. 5g and h).

To link cCREs to predicted target genes, we utilized scGLUE[24] and revealed that cCRE target genes had subclass-enriched expression

patterns (Fig. 2d). Motif enrichment analysis on region-specific cCREs with HOMER[25] identified candidate transcription factor (TF) regulators responsible for differentially accessible chromatin (Fig. 2e). To further profile regional specificity, we identified cCREs with region-specific accessibility enrichment within each subclass. In a uniformly down sampled subset of nuclei, IT projecting excitatory neurons and V1C uniqueness were again prominent drivers of regional variation (Fig. 2f). When normalized against the total number of peaks detected in each subclass, similar trends were observed (Supplementary Fig. 6a). Region-specific cCREs across all nuclei that passed filtering were identified by comparing accessibility in one region to all others

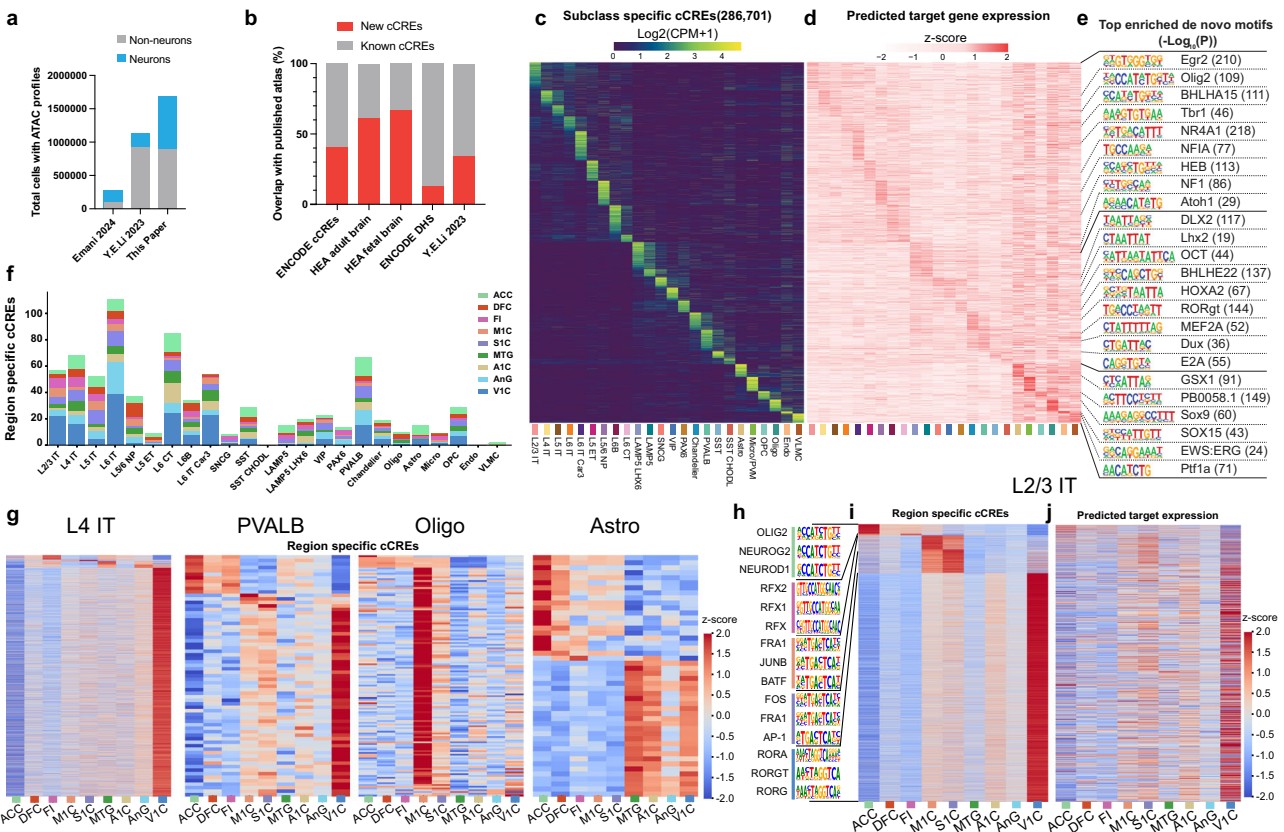

**Fig. 2 | Identification and regional specification of candidate cis-Regulatory Elements (cCREs) across human cortex. a** Bar plots showing the number of profiled nuclei that meet filtering cutoffs, neurons (blue) and non-neurons (gray), via single-cell ATAC-seq in this study and previously published resources. **b** Bar plots showing the percentage of new cCREs defined in this study (red) and overlapping cCREs with public resources (gray). **c** Heatmap showing chromatin accessibility of subclass enriched cCREs. **d** Heatmap of gene expression linked to subclass enriched putative enhancers as determined by scGLUE. **e** Top enriched de novo motifs for subclass-specific cCREs. Adjusted P-values calculated via one-sided hypergeometric test (BH corrected) without multiple testing correction. **f** Total counts of region-specific cCREs calculated as chromatin differentially accessible in one region as compared to all others combined, with uniform cell counts. Plots are organized by subclass and colored by region. **g** Heatmaps of region-specific differentially accessible cCREs for variable subclasses. Z-score calculated for each subclass. **h** Top enriched de novo motifs for L2/3 IT region-specific cCREs **i** Heatmap showing the chromatin accessibility of L2/3 IT regionally enriched cCREs. **j** Heatmap of gene expression linked to L2/3 IT regionally enriched putative enhancers determined by scGLUE. Source data are provided as a Source Data file.

(Fig. 2g, Supplementary Fig. 6b and Supplementary Data 4). Enriched transcription factor motifs and predicted target expression levels were identified using HOMER and scGLUE (Fig. 2h-j and Supplementary Fig. 6c).

In L2/3 IT neurons, regionally-distinct cCREs were enriched for unique TF motifs in each brain region (Fig. 2h), suggesting TF regulatory networks shape cortical regionalization. Utilizing cCREs, we performed linkage disequilibrium score regression[26] to assess regional enrichment of genetic heritability of DNA variants associated with various traits. Traits such as educational attainment and intelligence were significantly enriched in cCREs from IT neurons, as previously observed[10], but also had a distinct trend of enhanced enrichment for V1C-associated cCREs (Supplementary Fig. 6d).

**Spatial distribution of cellular subclasses in the human cortex**
The biological basis of regional neuronal specification remains poorly understood. While distinct transcriptional and chromatin patterns across cortical regions are established, it is unclear if these differences arise from changes in distal inputs, fundamental differences in GRNs, or effects of local cellular neighborhood composition.

To profile the spatial distribution of subclasses, we performed DART-FISH, a highly multiplexed spatial imaging technology[27], on tissue sections from each of the nine regions profiled via SNARE-seq2. We designed a 484-gene probe panel to detect and identify cellular

subclasses[28] (see methods and supplementary data 5). Following cell classification, we generated spatial maps for subclass distributions across nine tissue sections covering a total of 157,000 cells. (Fig. 3a and Supplementary Fig. 7a-c).

To enable comparison of spatial distribution across regions, we normalized cortical depth, spanning from the pial surface (defined by the absence of tissue beyond L2/3 IT neurons) to the white matter (identified by depletion of neuronal subclasses and significant oligodendrocyte enrichment). Once normalized, the regional density, defined as the estimated probability density function of each subclass, was calculated and compared across the cortical sheath (Fig. 3b and Supplementary Fig. 8a). This enabled identification of well-known changes across the cortical span, including the distinctive expansion of L4 in the visual cortex, observed as broadly distributed L4 IT neurons (Fig. 3b), and significant variation in astrocyte spatial distribution, consistent with prior reports[3,29] (Fig. 3b and Supplementary Fig. 8a).

Beyond laminar distribution, DART-FISH enabled the characterization of local cellular neighborhoods. Intricate communication amongst various cells within the brain is essential for proper brain functioning. While neurons extend projections that can connect with neighboring cells up to hundreds of millimeters away, astrocytes, oligodendrocytes, and microglia maintain extensive cell connections within hundreds of micrometers. To interrogate how the cellular composition of the microenvironment differs across regions, we

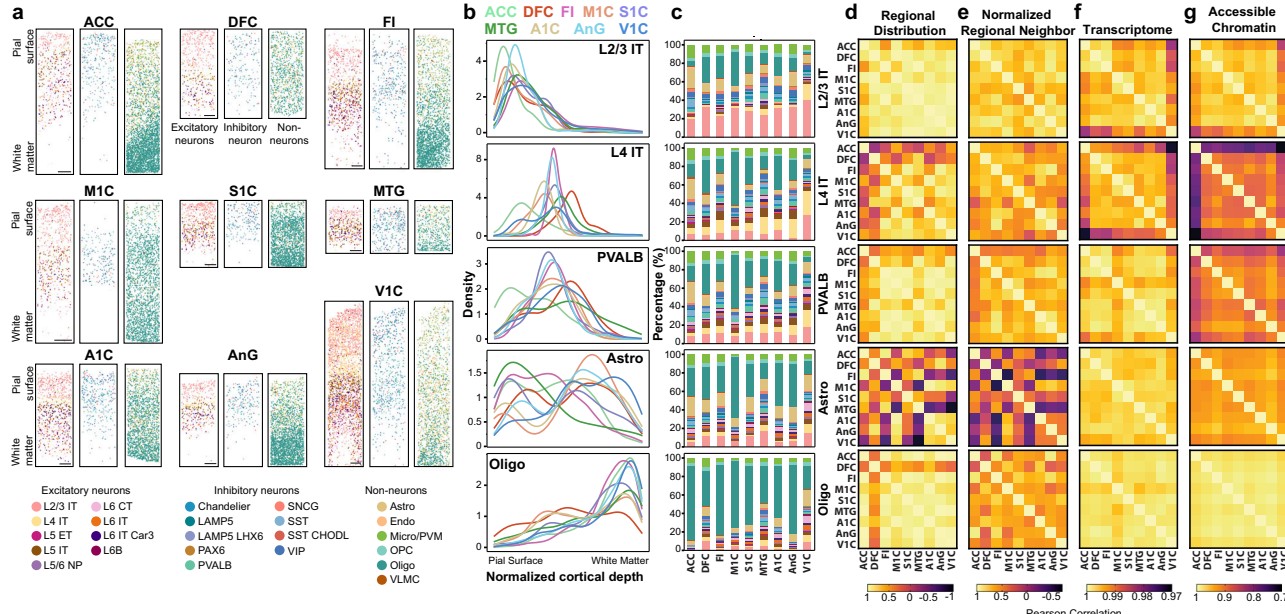

**Fig. 3 | Spatial distribution and regional variation of cell subclasses.**
**a** Visualization of called cell subclasses across the cortical sheet for each profiled region. Scale bars are 500μm. Regions of interest cropped from full tissue sections shown in fig. S7. Each region was profiled once via DART-FISH. **b** Subclass density across normalized cortical depth. For each section, markers of pial surface and white matter were used to establish the cortical sheet, and the density of each cell subclass was plotted as a density value across each region. **c** Stacked bar plots

visualizing neighborhood analysis for a specific cell subclass. Plotted values represent the percentage of called cells within 200μm of the primary cell subclass averaged across 100 profiled primary cells. **d**–**g** Pearson correlation across regions of regional density (**d**) normalized regional neighbors (**e**) transcriptomic gene expression (**f**) and chromatin accessibility (**g**) for specific cellular subclasses. Source data are provided as a Source Data file.

defined cellular neighborhoods. For a given cell subclass, we randomly sampled 100 cells per cortical region and quantified the subclass of all cells within a 200μm radius of each sampled cell. We then calculated the average proportion of each subclass within this neighborhood radius for each region (Fig. 3c, Supplementary Fig. 8a).

Regional differences in cellular cytoarchitecture were evident, with substantial shifts in the localization of specific cell subclasses and neighborhoods across certain regions (Fig. 3b and c). To interrogate how spatial differences relate to molecular variation in gene expression and chromatin accessibility, we computed Pearson correlations for each subclass across regions for estimated spatial distribution, neighbor composition, transcriptomic expression, and chromatin accessibility (Fig. 3d-g, see methods). Spatial correlation patterns were notably distinct from those based on transcriptome or accessibility data indicating possible limitations to the extent spatial variation drives subclass region-specific expression and accessibility.

## Gene expression and chromatin accessibility vary across the rostral-caudal (R-C) axis

The human neocortex exhibits distinct axes of functional variability. The R-C axis has been established as a key axis of variability that has major implications for regional functionality as well as disease progression and susceptibility. Many neurodegenerative diseases manifest by affecting discrete brain regions and progress via propagation along stereotyped trajectories[30–33]. Moreover, neurodevelopmental disorders, such as autism spectrum disorder (ASD), have been shown to affect disproportionally specific cortical areas[34]. Understanding how subclass-specific cellular changes are manifested across these functional axes could provide essential insight into the human brain.

The R-C axis across the neocortex (Fig. 4a) has well-characterized patterns of transcriptomic changes. Our transcriptomic data correlated closely with previous efforts profiling regional variability in gene expression along the axis[3] (Supplementary Fig. 9a). To systematically identify molecular variation across the axis, we determined what genes

and cCREs had activity that significantly correlated along the R-C axis (correlation for genes defined as absolute value Pearson correlation > 0.7 and adjusted p-value (BH corrected) <0.01, correlation for peaks defined as absolute value Pearson correlation > 0.5 and adjusted p-value (BH corrected) <0.05) (Fig. 4b and c and Supplementary Data 6). Less stringent cutoffs for peaks were used to account for the relative sparsity of ATAC input matrices. In line with the trends of region-specific variability, excitatory neurons had the largest number of R-C correlated genes and cCREs. To confirm that increased detection of correlated cCREs were not simply driven by differences in the total number of called peaks in each subclass, we normalized by total peak counts and found that L2/3 IT neurons not only had highest number, but also the highest fraction of R-C correlated cCREs (Supplementary Fig. 9b). The trend of excitatory neuron variability held when analyzing a uniformly down sampled subset of nuclei (Supplementary Fig. 9c). R-C correlated genes were typically subclass-specific with limited overlap between closely related subclasses (Supplementary Fig. 9d and e).

Single-nucleus multiomic profiling enabled the investigation of GRNs controlling axis variation. To identify and characterize the GRNs of the human neocortex, we utilized SCENIC +[35], which integrates chromatin accessibility and gene expression to infer enhancer- and TF-driven regulatory programs, termed eRegulons. As validation, we uncovered 182 subclass-enriched eRegulons composed of 12,361 genes and 67,472 cCREs (Supplementary Fig. 10a and Supplementary Data 7) with well-known subclass-specific TFs including SPI1 in microglia and OLIG2 in oligodendrocytes and oligodendrocyte precursors identified.

To profile regional variability of subclass regulatory networks in detail, we performed SCENIC+ analysis for subclasses with major axis variation, utilizing region-specific, differentially accessible chromatin regions for input. SCENIC+ analysis was performed on L2/3 IT, L4 IT, L5 IT, L6 IT, L6 CT, L5 ET, PVALB, VIP, and SST subclasses due to their elevated levels of axis correlated genes and peaks. For each subclass, SCENIC+ established eRegulons encompassing thousands of peaks

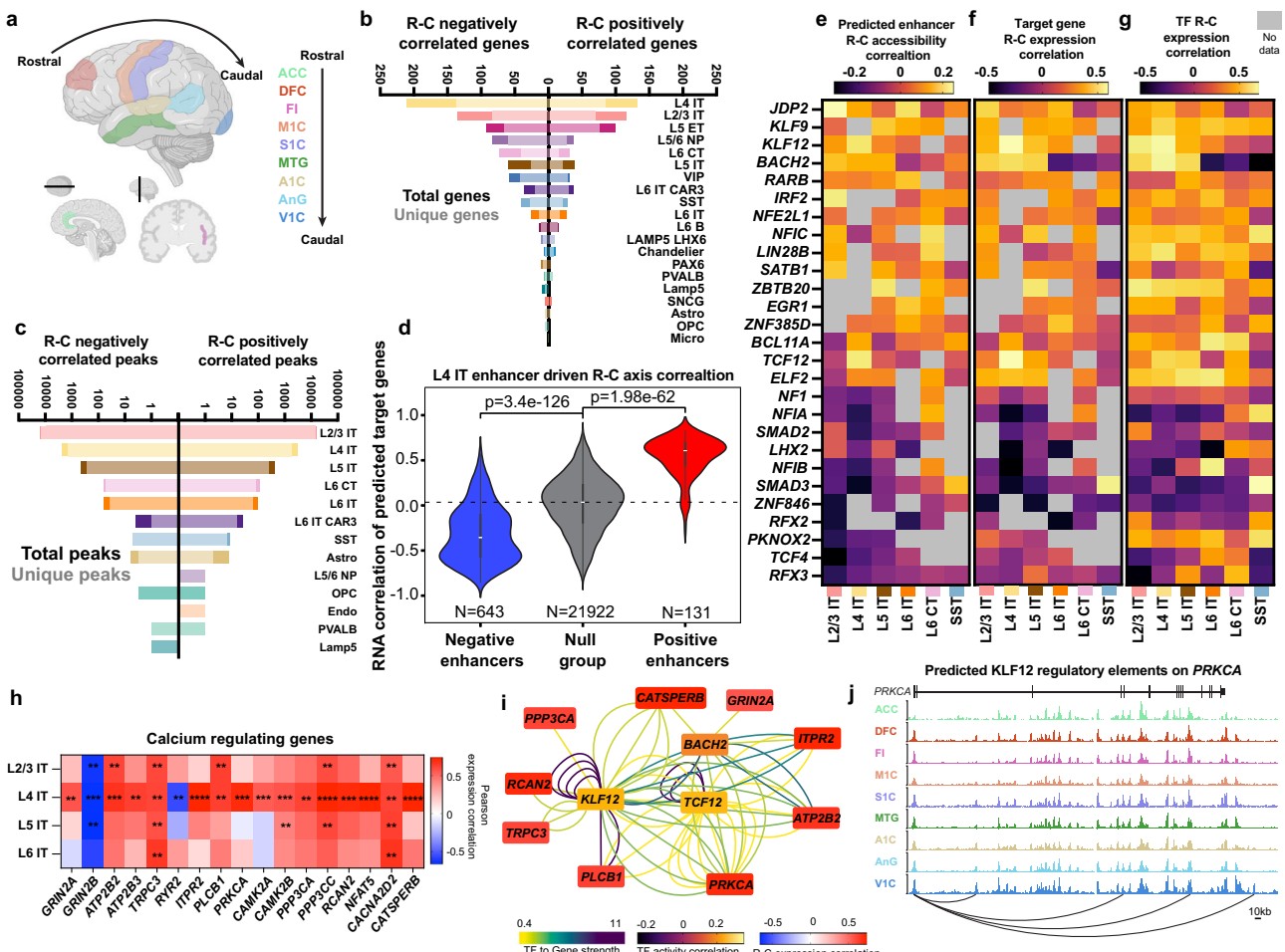

**Fig. 4 | Gene regulatory networks driving rostral-caudal (R-C) neuronal variation. a** R-C organization, created in BioRender. Costantino, I. (2026) https://BioRender.com/utnjof0. **b** Counts of total and unique genes correlated with the R-C axis. Unique genes are only correlated in specified subclass. Correlation is defined as Pearson correlation >0.7 and two-sided adjusted *p*-value (BH Corrected) < 0.01. **c** Counts of total and unique accessible chromatin peaks correlated with the rostral-caudal axis. Unique peaks are only correlated in specified subclass. Correlation is defined as Pearson correlation >0.5 and two-sided adjusted p-value (BH Corrected) < 0.05. **d** Transcriptomic correlation of predicted target genes from predicted enhancers identified as eRegulon components in L4 IT neurons. Enhancers were classified as positive (red), null (gray), or negative (blue), according to their R-C axis Pearson correlation. *p*-value from Mann-Whitney U Test. (Positive enhancers: minima −0.85, maxima 0.59, mean 0.−0.32, bottom box 25th percentile bound −0.57, upper box 75th percentile bound −0.11; Null: minima −0.85, maxima 0.83, mean 0.01, bottom box 25th percentile bound −0.19, upper box 75th percentile bound 0.23; Negative enhancers: minima −0.25, maxima 0.82, mean 0.54, bottom

box 25th percentile bound 0.44, upper box 75th percentile bound 0.69. **e** Transcription factor activity correlation across R-C axis. Values are the average Pearson correlation of predicted binding sites for specified TFs from SCENIC+ analysis. **f** Target gene expression correlation across R-C axis. Values are the average Pearson correlation of expression of target genes for specified TFs from SCENIC+ analysis. **g** Transcription factor expression Pearson correlation across the R-C axis. **h** Pearson correlation across R-C axis of transcriptomic expression of genes associated with regulation of intracellular calcium levels. Asterisks denote two-sided adjusted *p* value (BH corrected) (**$p < 0.01$, *** $p < 0.001$, **** $p < 0.0001$). **i** Gene regulatory network of calcium-regulating genes in L4 IT projecting neurons. Lines represent enhancer-gene-transcription factor connections predicted by SCENIC +. Colored by transcription factor to gene strength. Transcription factors (central nodes) are colored by activity correlation across the R-C axis as defined in (**e**). Calcium genes colored by R-C expression correlation. **j** Epigenome tracks for KLF12 predicted binding to *PRKCA*. Source data are provided as a Source Data file.

and genes as well as nearly 100 TFs, offering broad insight into cellular regulatory control (Supplementary Fig. 10b and Supplementary Data 8). Enhancers with positively correlated R-C accessibility predominantly regulated genes that had positive R-C correlated expression in L4 IT neurons (Fig. 4d) and additional subclasses (Supplementary Fig. 10c). To confirm the inverse, genes with positively correlated R-C accessibility showed predicted enhancers with R-C correlated accessibility (Supplementary Fig. 10d). The same held true for negative correlations (Fig. 4d Supplementary Fig. 10c and d).

We identified TFs with activity (defined as the accessibility of their predicted binding sites) that varied along the R-C axis. Several TFs were identified with R-C correlated activity (Supplementary Data 9), several of which were responsible for correlated activity in two or more subclasses (Fig. 4e). Critically, the expression of the target genes of the

predicted enhancers regulated by these TFs (Fig. 4f) as well as the expression of these TFs themselves (Fig. 4g) predominantly matched the correlation directionality of their defined activity.

## Gene regulatory network (GRN) controlling the R-C axis of IT projecting neuron calcium regulation

To profile the functional pathways associated with gene expression changes along the R-C axis, we performed gene set enrichment analysis on positively and negatively correlated genes for each cell subclass (Supplementary Fig. 11a). For L4 IT neurons, the only significantly enriched terms for positively correlated genes related to aspects of intracellular calcium regulation and signaling (Supplementary Fig. 11b). Further interrogation revealed numerous genes critical for controlling both intracellular calcium levels and calcium signal transduction were

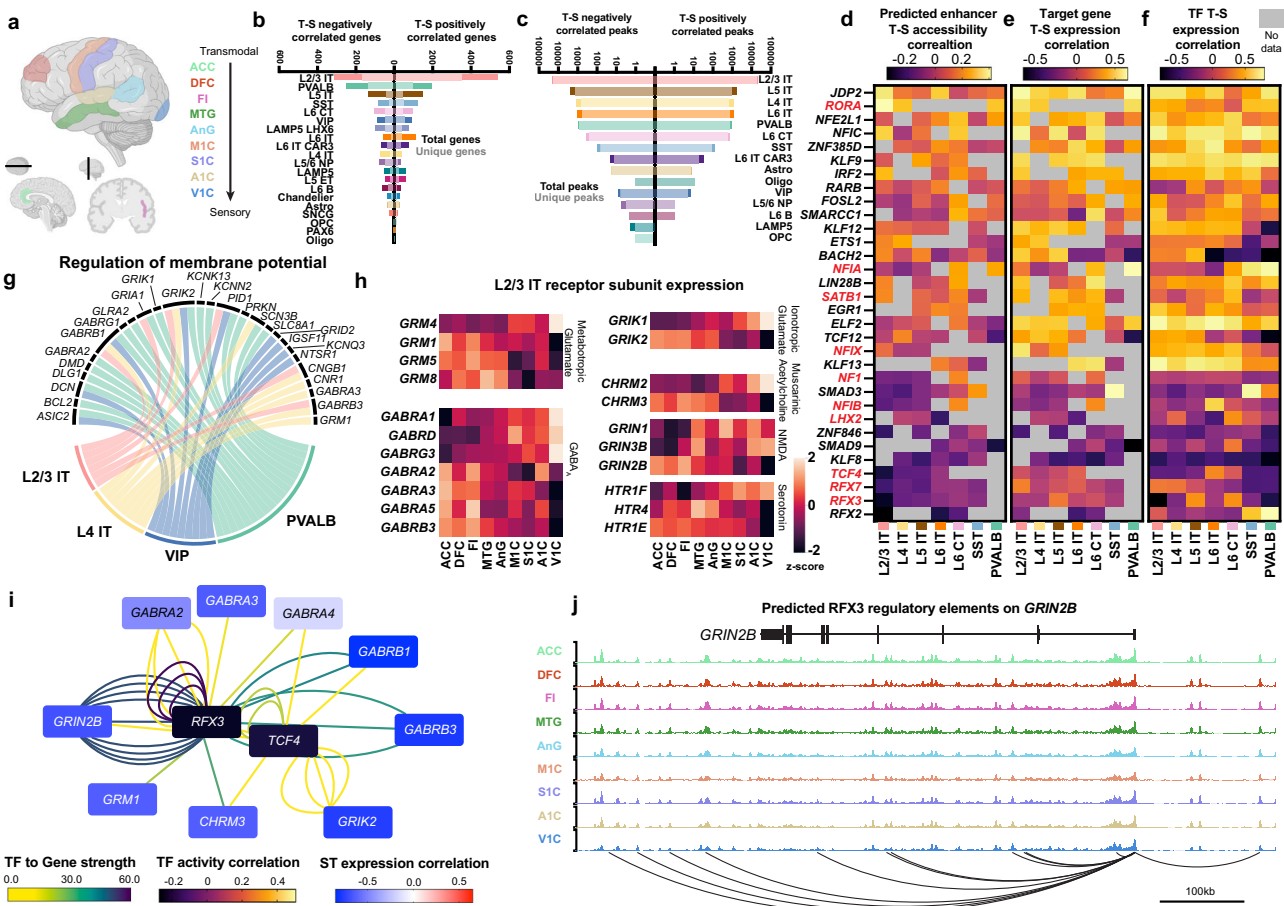

**Fig. 5 | Gene regulatory networks responsible for receptor subunit switching. a** Transmodal to sensory (T-S) regional organization, Created in BioRender. Costantino, I. (2026) https://BioRender.com/utnjof0. **b** Counts of total and unique genes positively or negatively correlated with the T-S axis. Unique genes are only correlated in specified subclass. Correlation is defined as Pearson correlation >0.7 and two-sided adjusted p-value (BH Corrected) < 0.01. **c** Counts of total and unique accessible chromatin peaks positively or negatively correlated with the rostral-caudal axis. Unique peaks are only correlated in specified subclass. Correlation is defined as Pearson correlation >0.5 and two-sided adjusted p-value (BH Corrected) < 0.05. **d** Transcription factor activity correlation across T-S axis. Values are the average Pearson correlation of predicted binding sites for specified TFs from SCENIC+ analysis. TFs in red are identified in SFARI database. **e** S-T correlation of transcriptomic expression of target genes of the predicted enhancers profiled in

(**d**). Value is the average Pearson correlation of all predicted target genes. **f** Transcription factor expression Pearson correlation across the T-S axis. **g** Genes associated with the term regulation of membrane potential and cell subclasses identified via gene set enrichment analysis of genes with expression positively correlated with the T-S axis. **h** T-S expression Z-scores of various receptor subunits in L2/3 IT neurons by region. **i** Gene regulatory network of T-S negatively correlated receptor subunit genes in L2/3 IT neurons. Lines represent enhancer-gene-transcription factor connections predicted by SCENIC+. Lines are colored by transcription factor to gene strength. Transcription factors (central nodes) are colored by activity correlation across the T-S axis as defined in (**d**). Calcium genes are colored by T-S expression correlation. **j** Epigenome tracks for RFX3 predicted binding to *GRIN2B*. Source data are provided as a Source Data file.

significantly correlated across the R-C axis in IT projection neurons, especially those of Layer 4. These include modulators of intracellular calcium levels such as NMDA receptors (*GRIN2A* and *GRIN2B)*, PMCA pumps (*ATP2B2* and *ATP2B3*), the TRP receptor (*TRPC3*), endoplasmic reticulum channels IP3R (*ITPR2*) and RYR (*RYR2*) as well as downstream effectors including the calcium sensitive kinases PKC (*PRKCA*) and CAMKII (*CAMK2A* and *CAMK2B*) (Fig. 4h and Supplementary Fig. 11c). In Alzheimer's disease, neuronal subtypes show increased susceptibility to death in the cortex[36,37]. RORB-expressing L4 IT neurons[36] and L2/3 IT neurons[37] have been identified as the first neurons to die during disease progression. Calcium dysregulation can contribute to disease progression in AD[38], and a shift in NMDA receptor subunit composition towards GluN2B (encoded by *GRIN2B*) has been shown to display slower kinetics as compared to GluN2A with important implication in long-term potentiation and neuronal signaling[39,40] implicating important functional roles of the observed changes to calcium regulation.

To identify potential upstream regulators of this calcium module, we profiled which TFs regulated the expression of these calcium pathway genes via SCENIC+. Of the TFs we identified, we focused on

those that had activity (Supplementary Fig. 11d) and expression (Supplementary Fig. 11e) correlated across the R-C axis. BACH2, KLF12, and TCF12 formed a GRN responsible for a distinct pattern of increased expression of calcium signaling genes across the R-C axis in L4 IT projecting neurons (Fig. 4i) with distinct regulation of calcium activated kinases, like *PRKCA* by KLF12 (Fig. 4j).

## Receptor subunit switching across the transmodal-sensory (T-S) axis

The T-S axis separates cortical regions based on their functional roles, distinguishing sensorimotor regions (motor, somatosensory, auditory, and visual cortices) from transmodal regions, responsible for integrating diverse sensory and subcortical inputs (Fig. 5a)[13,41]. While both the T-S and the R-C axes are anchored by ACC and V1C regions at either end, the rearrangement of the other regions revealed distinct correlated genes, accessible cCREs, and functional pathways.

While L2/3 IT neurons had the largest number of genes and cCREs correlated along the T-S axis, inhibitory subclasses, especially MGE-derived interneurons including PVALB and SST, had more axis-

correlated genes than other IT projecting excitatory neurons when all nuclei were profiled (Fig. 5b and c and Supplementary Data 10) and when analysis was performed on uniformly down sampled data (Supplementary Fig. 12a). Axis-correlated genes were again often unique to one subclass (Supplementary Fig. 12b and c). Importantly, only a limited number of genes were determined to be both positively correlated, or both negatively correlated across the R-C and T-S axes (Supplementary Fig. 12d), underscoring the importance in the organization of regions beyond ACC and V1C in determining correlation. Similarly to the R-C axis, SCENIC+ enhancer-target gene pairs showed matching regional variance, both in correlated genes and their associated enhancers (Supplementary Fig. 12e) as well as correlated enhancers and their target genes (Supplementary Fig. 12f).

Axis-correlated genes were again controlled by GRNs that themselves had correlated activity (Fig. 5d and Supplementary Data 11), expression of predicted target genes (Fig. 5e), and TF expression (Fig. 5f). While there was conservation of TFs such as JDP2, RORA, and RFX3 in driving variation across both the R-C and T-S axes, there were also TFs such as NFIA in L2/3 neurons that switch from negative (R-C) to positive (T-S) correlated activity. To better understand the functional consequences of T-S axis-correlated genes, we performed gene set enrichment analyses separately on the positively and negatively correlated genes for each subclass. Enriched terms were observed across various subclasses, including regulation of membrane potential and synapse assembly (Fig. 5g and Supplementary Fig. 13a).

Closer examination revealed that many genes that correlated along the axis (both positively and negatively) encoded subunits of major neuronal signaling receptors (Supplementary Fig. 13b). Often, specific subunits increased in expression along the T-S axis, while different subunits for the same receptor decreased. In L2/3 IT neurons, this dynamic expression was particularly evident, with subunits of metabotropic and ionotropic glutamate, NMDA, muscarinic acetylcholine, and serotonin receptor switching. The largest number of subunits changed for $GABA_A$ receptors (Fig. 5h). While changes along cortical axes in protein levels and activity of these receptors have been previously reported[14,42], the observation that specific subunits of the receptors are switching adds an additional level of complexity to neuronal receptor variability in the human cortex.

To probe upstream regulation of this receptor subunit remodeling, we profiled SCENIC+ TFs responsible for both positively and negatively correlated receptor subunit genes (Supplementary Fig. 14a and b). Negatively correlated receptor genes were strongly regulated by RFX3 and TCF4 (Fig. 5i), both of which displayed significant expression correlation with the T-S axis (Supplementary Fig. 14c). RFX3 was predicted to bind eight separate sites of the NMDA receptor subunit gene GRIN2B (Fig. 5j). Genetic variants of GRIN2B result in neurodevelopmental disorders[43–45]. Point mutations in RFX3 have been shown to cause neurodevelopmental changes resembling ASD[46], and RFX3 has been implicated in distinct roles in neurodevelopment[47]. Mutations to TCF4 cause the neurodevelopmental disorder Pitt-Hopkins syndrome, which is also characterized by ASD-like behavioral changes[48]. To further probe the role of axis-correlated TFs in autism, we screened the 32 identified T-S correlated TFs against the SFARI database[49] and found that 10 of the 32 genes are implicated in driving ASD (Fig. 5d), with further enrichment in TFs with negatively correlated axis activity. These findings raise the possibility that dysregulated NMDA subunit levels throughout the cortex, controlled by TCF4 and RFX3 axis-specific activity, may contribute to neurological changes that control the distinct regional effects observed in ASD brains[34].

## Discussion

The human brain possesses complex regionalization and distinct interconnectedness, which underlie its function. In this work, we performed a deep single-cell multiomic analysis of the human cerebral cortex to profile the molecular changes to cell subclasses across nine regions of the cortex. By integrating these datasets, we identified the transcription factors (TFs) and gene regulatory networks (GRNs) responsible for distinct molecular changes that enable aspects of regionalized functionality. We confirmed that differences in cortical profiles occur primarily in neurons, especially intratelencephalic projection neurons, with strikingly few changes in non-neuronal populations. Genes that changed in expression and chromatin accessibility across the cortex distinctly centered on regulation of neuronal communication and downstream signal transduction. Careful spatial profiling of these cellular subclasses revealed that regional spatial variation did not align with changes associated with expression or chromatin accessibility. GRNs were identified in which both the expression and activity of distinct TFs correlated across defined axes within the human brain and regulated gene expression in a matching manner. These TFs appear to play essential roles in the functional changes of IT projecting neurons across the cortex and may control differences in neuronal firing via receptor subunit switching as well as information transfer and consolidation as molded by calcium signaling.

Regions of the human cortex differ at a cellular and molecular level to account for the functional responsibilities of each region. While such differences have been broadly characterized[3], the precise mechanisms neurons use to adapt their behavior to account for regional functional specialization, and the transcriptional regulatory programs enabling this adaptation have remained incompletely understood. The clear rostral-caudal spatial pattern of calcium regulation in IT projecting neurons identified here suggests region-specific control of calcium mediated network dynamics and system-level consolidation. While distinct gradients of neuronal receptors have been observed across the cortex[14], our finding that the subunit composition may also vary adds an additional layer of regionalized firing properties capable of distinguishing functional responsibilities. The identification of the RFX family of transcription factors as key regulators of this depolarization control offers additional insight into their capability to fine-tune neuronal firing and highlights how mutations may lead to neurodevelopmental alterations to circuit dynamics as seen in autism spectrum disorder. While our findings are consistent with previous transcriptomic and epigenomic atlases[3,10], they expand beyond these efforts by providing deep insight into the molecular and regulatory elements that drive neuronal regionality. Additionally, future studies could benefit from spatially mapping these changes directly onto human brain atlases, enabling a more comprehensive and quantitative analysis of spatial gradients across cortical regions.

While this work provides insights into the spatial distributions and transcriptomic and epigenomic changes across the human cortex that enabled the identification of TFs driving regional specification, it begs the question of what aspects of brain biology drive these molecular differences themselves. Two possibilities are intra-region neuronal connectivity and development. It is well-known that different cortical regions receive distinct inputs, from variable intracortical and deep sub-cortical regions of the brain. Additionally, brain development is complex and highly regulated. While a cursory interrogation of overlapping region-specific DEGs in the adult and developing brain identified overlapping transcription factors, a more comprehensive integration of the data presented here with projection and developmental information could reveal integrative mechanisms underlying the fundamental underpinnings of cortical regionality.

The anticipated neuronal heterogeneity in the cerebral cortex was driven by distinct TFs sculpting chromatin and transcriptomic landscapes. These regional changes may underlie both the progressive neuroanatomical changes seen in neurodegenerative disorders like Alzheimer's disease as well as cortical changes seen in

neurodevelopmental disorders like autism and schizophrenia, where cortical regional variation has been observed. The identified regulation of subunit switching offers potentially novel targets towards developing therapeutic treatment of human brain disorders. Broadly, this work establishes a foundational understanding of how epigenomic diversity across cortical regions influences specialized cellular functions. The identification of gene regulatory networks controlling spatial organization of essential neuronal pathways offers insight into the molecular basis of cortical specialization.

## Methods

### Cohort selection

Postmortem brain tissue and donor metadata were obtained via the University of Washington BioRepository and Integrated Neuropathology (BRaIN) laboratory. In compliance with all ethical standards, informed consent for research brain donation was obtained according to protocols approved by the University of Washington Institutional Review Board and/or the Uniform Anatomical Gift Act. The study cohort was selected from available donor brains with the required brain regions, uniform tissue collection and processing procedures, and without major neuropathological features indicative of neurodegenerative diseases. No randomization was applied during cohort selection. Only one female donor met these inclusion criteria, resulting in an unequal distribution of male and female donors in the study. Detailed donor characteristics are provided in Supplementary Data 1.

### Tissue preparation and nuclei isolation

Appropriate tissue sections were carefully identified and sections spanning from the gray matter to pial surface were cut from larger tissue blocks. All tissue was profiled for RNA integrity number (RIN) and only sections with a RIN > 7 were used. Frozen tissue sections were incubated in 1 ml of cold nuclei isolation buffer [320 mM sucrose, 100 μM EDTA, 1 mM DTT, 10 mM Tris-HCl pH 8, 5 mM $CaCl_2$, 3 mM $MgAc_2$, 0.1% Triton X-100, 160U/ml RNase inhibitor (Takara Bio #2313 A), cOmplete EDTA-free protease inhibitors (Roche #11836170001)] for 15 min and homogenized using a tissue grinder (Wheaton #358005) 10–15 times to mechanically dissociate the tissue. The homogenate was passed through a 50 μm filter to remove debris and centrifuged at 820 g for 5 min at 4 °C. The isolated nuclei were washed twice with wash buffer + sucrose (PBS with 320 mM sucrose, 250 μM EGTA, 1 mM DTT, 40U/ml RNase inhibitor, protease inhibitors). Nuclei were then fixed my resuspending the pellet in 5 ml of cold 0.5% formaldehyde in 1X PBS. Nuclei were wash buffer + BSA (PBS with 1% BSA, 250 μM EGTA, 1 mM DTT, 40U/ml RNase inhibitor, protease inhibitors). Nuclei were stained with 1.25 μg/ml DAPI (Sigma #10236276001) diluted in the wash buffer plus BSA for 5 min. The nuclei were sorted on a FACSAria Fusion (BD Biosciences) by gating out debris using the forward scatter and side scatter plots and selecting DAPI+ singlets.

### Nissl staining

Brain sections (20 μm) were mounted on glass slides [Fisherbrand™ Tissue Path Superfrost™ Plus Gold Slides, 22-035813] and fixed in 10% neutral buffered formalin (NBF) [Epredia, 22-050-104] for 5 minutes at room temperature. Sections were incubated in 70% and 95% ethanol for 3 minutes each, then 100% ethanol [Decon™ Labs, 04-355-222] for 20 minutes, followed by rehydration steps in 95% and 70% ethanol, before a quick rinse in deionized (DI) water. Sections were incubated in 0.2 % Cresyl violet solution for 5 minutes. Excess stain was removed with DI water (3 × 30 second washes), 70% ethanol and 95% ethanol for 1 minute each, and xylene [Fisher Chemical, X5SK-4] for 3 minutes. Coverslips were used with Cytoseal 60 mounting medium [Epredia, 23-244257] to seal the slides. Slides were then imaged using a Keyence BZ-X810 microscope.

### Library preparation and sequencing

SNARE-seq2 experiments were performed similar to previous publication[15], and a detailed step-by-step protocol is available at https://www.protocols.io/view/snare-seq2-kqdg35yeqv25/v1. Briefly, 1-2 million nuclei were resuspended with 1x PBS + RNase inhibitors (0.05 U/μL SUPERase In [ThermoFisher, Cat# AM2696], 0.05 U/μL RNaseOUT [ThermoFisher, Cat# 10777019]) and kept on ice. 1 mL of pre-chilled 1% formaldehyde was added to nuclei suspension, gently mixed with pipette, and incubated on ice for 10 mins. Nuclei were spun-down at 900 g for 6 mins at 4 °C and resuspended in 1X PBS containing 0.1% BSA and RNase inhibitors. Nuclei were spun-down again at 900 g for 8 mins at 4 °C and resuspend in 1X Tango Buffer (ThermoFisher, Cat# BY5) with 16% DMF (Sigma, Cat# 100397) and RNase inhibitors to a concentration of 3,400 nuclei/μL. Tagmentation reactions were performed in 1X Tango buffer with 16% DMF and RNase inhibitors for 30 mins, 300 rpm at 37 °C. After washing, 8,000 nuclei were aliquoted into 48 wells. Nuclei were labeled with Round 1 AC oligo by T7 DNA ligase, followed by reverse transcription with Maxima H Minus Reverse Transcriptase (EP0751). Next, nuclei were further barcoded by two rounds of ligation-based combinatorial barcoding by T4 DNA Ligase (NEB, Cat# M0202L). After lysis by Proteinase K (NEB, Cat# P8107S), cDNA and DNA fragments were purified with streptavidin beads (ThermoFisher, Cat# 65001) and template switching was then performed on cDNA. Both cDNA and DNA were co-amplified in the first-round PCR, followed by splitting into DNA and RNA libraries for amplification of sequencing libraries. Libraries were sequenced with NovaSeq S4. DNA libraries were sequenced using a 300 cycle kit (Read1, 75 bp; Index1, 94 bp; Index2, 8 bp; Read 2, 75 bp). RNA libraries were sequenced using a 200 cycle kit (Read1, 70 bp; Index1, 6 bp; Read2, 102 bp). SNARE-seq2 data from a previous publication[2] from a single donor and only M1C, were incorporated to increase depth of profiling for the M1C region.

### Preprocessing and read alignment

Data preprocessing was performed as in the previous publication (15). Briefly, RNA reads were mapped to GRCh38 reference genome with STAR (v2.5.2b), and DNA reads were mapped to the reference genome with Snaptools v1.4.7. Quality filtering of cell barcode was first performed on RNA counts to exclude cells with fewer than 200 genes. Each cell barcode contains library batch identification code, and all samples were combined across experiments for downstream analysis. Cell barcodes were further filtered based on DNA profiles to remove cells with fewer than 500 fragments for non-neuronal cells or fewer than 1,000 fragments for neuronal cells as well as cells with TSS < 1.

### Clustering and subclass annotation

Cell clustering based on RNA profile was performed with Seurat V5. The cell-to-gene counts matrix was normalized with the SCTransform function, and 2,000 variable genes were selected for dimension reduction with PCA and clustering with the Leiden algorithm. Possible doublets and low-quality clusters were manually removed according to lack of overrepresented marker genes or having multiple marker genes across different major cell types. Annotation of cell clusters was based on known marker genes from prior study[3].

### edgeR differential gene expression analysis

We used edgeR (version 4.6.0) to identify differentially expressed genes (DEG) and region-specific cCREs across brain regions for a given cell type. We combined reads for each gene from all cells per donor and brain region and used the resulting gene by group matrix as the input to the edgeR quasi-likelihood negative binomial generalized loglinear model (glmQLFit). Each donor was treated as a replicate for a brain region in the model. We removed lowly expressed genes with the edgeR function filterByExpr by setting min.count to 10 and min.total.count to 15. Similarly, we removed peaks with low coverage with

filterByExpr by setting min.count to 5 and min.total.count to 15. We used the contrast comparing one region to the average of all the other regions to identify DEGs/ region-specific cCREs. After fitting the model, we used the statistical significance cutoff of FDR < 0.01 for DEGs and FDR < 0.05 for region-specific cCREs respectively. We used the resulting DEGs for the gene set enrichment analysis.

To additionally compare the number of DEGs across different subclasses, we down sampled the number of cells to 1,000 in each of the 9 brain regions for each subclass so that the effect of read depth and sampling variations on the comparisons are minimized. We used the same procedure described above for the DEG analysis with the down sampled dataset.

To account for the fact that the number of expressed genes in a given cell subclass could affect the power for DEG analysis, we normalized the number of DEGs in each region for each cell type to the number of genes after filtering on expression levels, as outlined above (filterByExpr).

### MAST differential gene expression analysis

We used the MAST method in FindMarkersAll from Seurat (5.1) to identify marker genes for each region in each cell type. We included CDR (cellular detection rate), sex, age, PMI and RIN as covariates when fitting the MAST model. We kept all other parameters were kept at their default settings. Genes with an p_val_adj <0.001 and avg_log2fc > 0.5 were considered differentially expressed genes (DEGs).

### Differential accessible region analysis

To identify differentially accessible regions, single-cell chromatin accessibility signals were first aggregated for each cell type according to RNA-based clustering and annotation. Peak calling was performed with MACS2 for each cell type and merged to obtain the union peak set (as in [10]). Next, reads counts for each peak were calculated for identification of region-specific cCREs that were significantly enriched with each cell grouping against a selection of background cells of matched total peak numbers. Prediction of target genes was performed with scGLUE, and significant regulatory connections with Q-value < 0.05 were retained. Transcriptional factors motif enrichment analysis was performed on region-specific cCREs with HOMER software[25].

We used the same down sampling approach described in the last section. Due to the high sparsity of the scATAC count matrix, we used p-value < 0.05 as statistical cutoff for identifying differential accessible regions for down sampled comparisons. To account for the fact that the number of peaks in each cell subclass could affect the power for region-specific cCREs analysis, we normalized the number of region-specific cCREs in each region for each subclass to the number of peaks called in that subclass.

### ScGLUE cCRE target gene prediction

Prediction of target genes for cCREs was performed using the scGLUE[24]. For the scRNA-seq data, highly variable genes were selected with the "scglue.models.configure_dataset" function, specifying a Negative Binomial ("NB") distribution and PCA-based representation. Similarly, for scATAC-seq data highly variable peaks were selected using the same "NB" distribution and based on LSI-based representation. A guidance graph was constructed based on the subgraph of highly variable features and the "PairedSCGLUEModel" was then trained via the "scglue.models.fit_SCGLUE" function.

### Motif enrichment analysis for region-specific cCREs across brain regions

We used HOMER2 (version 4.11) to identify enriched transcription factor binding motifs from the set of region-specific cCREs across brain regions. For a given subclass, the background peak set for the enrichment analysis for a given brain region was constructed by filtering out the region-specific cCREs from the subclass-specific peaks. We used findMotifsGenome.pl from the HOMER2 toolset with the following parameters: -size given, -mask, -bg background set. We utilized the HOMER2 known transcription factors database to extract the outputs.

### Gene set enrichment analysis

We used clusterProfiler (version 4.16.0) for gene set enrichment analysis (GSEA). In each GSEA analysis, we used edgeR filterByExpr (min.count to 10 and min.total.count to 15) to remove lowly expressed genes across regions and used the filtered genes as the background set. We used the enrichGO function from clusterProfiler for the GSEA analysis by setting the ontology database to "BP" (Biological Processes) and qvalueCutoffs to 0.1. The simplify function is used to remove redundant Gene Ontology terms from the raw enrichGO output.

### LDSC analysis

We used https://github.com/bulik/ldsc for the LDSC analysis[26]. The GWAS summary statistics for LDSC analysis are prepared for quantitative traits related to neurological disease and control traits with summary statistics organized into standard format for Linkage disequilibrium regression analysis. We downloaded VCF dataset of EUR ancestry individuals from 1000 Genome Project Phase 3 release and use the software plink to prepare the bim/bed/fam files as the reference variant set for the LDSC analysis. We downloaded the summary statistics from the public GWAS datasets and use munge_sumstats.py from ldsc package to convert to the standard format for LD regression analysis[10]. We called peaks for each brain region separately in each subclass and used the consensus peak set across all regions and all subclasses as the background set for the linkage disequilibrium (LD) regression analysis.

### DART-FISH gene selection

A 484-gene panel was selected with gpsFISH[28]. SNARE-seq2 datasets from the human cortical samples with cell type annotation at subclass level 2 were used as input of gpsFISH. Additional curated marker genes were incorporated from prior knowledge to uncover 24 subclasses across excitatory neurons, inhibitory neurons, and non-neuronal cells.

### DART-FISH probe design

The constitutive exons were defined as regions in RefSeq where at least 33% of isoforms overlap. We used a modified version of ppDesigner (https://github.com/Kiiaan/sppDesigner) to identify padlock target sequences along the constitutive exons for target genes. ppDesigner was run on the setting: no overlap between probes allowed. Individual arms were constrained between 17nt and 22nt long with the total target sequences no longer than 40nt. The resulting target sequences were aligned to GRCh38/hg38 with BWA-MEM[50] and sequences with MAPQ < 40 or secondary alignment were removed. We further removed probes that have GATC (DpnII recognition site). A maximum of 40 probes per gene were selected. Finally, the target sequences were concatenated with amplification primer sequences, universal sequence, and gene-specific decoder sequences to produce final padlock probe sequences and were ordered as an oligo pool from Twist Bioscience (South San Francisco, CA). The 4-on-4-off scheme with Hamming distance > =2 was used to encode the genes. The sequences of each padlock probe can be found in data S5. The brain probe set was amplified with the AP1V6U and AP2V6 primer pair. Probes targeting PLP1 gene were added subsequently during padlock probe capture in the DART-FISH experiment (see details below). The sequences of PLP1 probes can be found in data S5.

### Padlock probe production

The seed oligo pool was amplified with AP1V7U and AP2V7 primers and purified with QIAquick PCR purification kit. The purified PCR product

was quantified with Qubit HD dsDNA kit and normalized to 10 nM as "10 nM oligo pool 1st amplicon". In padlock probe mass production, the 10 nM oligo 1st amplicon was further amplified with AP1V7U and AP2V7 primers and purified with ethanol precipitation. The uracil-tagged strand of the dsDNA amplicon was digested with lambda Exo. USER and Dpn II enzymes were later used to cleave the PCR amplification handles at 5' and 3' end. Afterwards, TBU gel electrophoresis and size selection were carried out to select the fully cleaved ssDNA and ethanol precipitation was used to purify the final padlock probes for the DART-FISH experiments. Additional details can be found on protocols.io (dx.doi.org/10.17504/protocols.io.n92ldm3pxl5b/v1).

## DART-FISH reverse transcription and cDNA crosslinking

Tissue sections were fixed in 4% PFA in 1x PBS at 4 °C for 1 h, followed by one 3-minute wash with DEPC-PBST (1x PBS and 0.1% Tween-20) at 4 °C and one 3-minute wash with DEPC-PBST at room temperature. Then, a series of 50%, 70%, 100%, and 100% ethanol were used to dehydrate the tissue sections at room temperature for 5 min each. Tissues were then air dried for 5 min and silicone isolators (Grace Bio-Labs, 664304) were attached around the tissue sections. Tissue sections were permeabilized with 0.25% Triton X-100 in DEPC-PBS (1x PBS) at room temperature for 10 min, followed by two 1 mL DEPC-PBS (1x PBS) washes and a 1 mL DEPC-water wash. Next, the sections were digested with 0.01% pepsin in 0.1 N HCl (pre-warmed 37 °C for 5 min) at 37 °C for 90 s and washed with 1 mL DEPC-PBS (1x PBS) twice. Afterwards, dT20 and N9 primers (dT20_dc7-AF488 and N5_dc10-Cy5_N9) were mixed to a final concentration of 2.5 µM with the reverse-transcription mix (10U/µL SuperScript IV (SSIV) reverse transcriptase, 1x SSIV buffer, 250 µM dNTP, 40 µM aminoallyl-dUTP, 5 mM DTT, 0.4U/ µL Superase-In RNase inhibitor). The sections with the mix were incubated at 4 °C for 10 min and then transferred to a humidified 37 °C oven for overnight incubation. After reverse transcription, tissue sections were washed with 1xPBS twice and incubated in 5 mM BS(PEG)₉ in 1x PBS at room temperature for 1 h. The tissue sections were washed twice with 1x PBS, followed by incubation with 1 M Trish, pH 8.0 at room temperature for 30 min. Subsequently, the sample was washed with 1 mL 1xPBS twice resulting in cDNA crosslinked to the tissue section. Additional details can be found at protocols.io (dx.doi.org/10.17504/protocols.io.yxmvmnd45g3p/v1).

## Padlock probe capture

After cDNA crosslinking in tissue section, remaining RNA was digested with RNase mix (0.25U/µL RNase H, 1% Invitrogen RNase cocktail mix, 1x RNase H buffer) at 37 °C for 1 h followed by two 1x PBS washes. The padlock probe library was mixed with the Ampligase buffer. Then, the mixture was heated to 85 °C for 3 min and cooled on ice. Subsequently, the mixture was supplemented with 0.33 U/µL Ampligase enzyme such that the final concentration of the padlock probe library was 100 nM for the brain probe set, in 1x Ampligase buffer. Finally, the samples were incubated with probes at 37 °C for 30 min, and then moved to a 55 °C humidified oven for overnight incubation.

## RCA and rolony crosslinking

After padlock probe capture, the tissue sections were washed with 1 mL 1x PBS twice and hybridized with RCA primer solution (0.5 µM FISSEQ_ppRCA primer, 2x SSC, and 30% formamide) at 37 °C for 1 h. Then, the tissue sections were washed with 1 mL 2x SSC twice and incubated with Phi29 polymerase solution (0.2 U/µL Phi29 polymerase, 1x Phi29 polymerase buffer, 0.04 mM aminoallyl-dUTP, 1 mg/mL BSA, and 0.25 mM dNTP) at 30 °C in a humidified chamber for 5 h. Afterwards, the tissue sections were washed with 1 mL 1x PBS twice and the rolonies were crosslinked and embedded in polyacrylamide gel. Specifically, the tissue sections were incubated in 0.2 mg/mL Acryloyl-X, SE in 1x PBS at room temperature for 30 min. Then, the tissue sections were washed once with 1 mL 1xPBS, followed by incubation with

4% acrylamide solution (acrylamide/bis 37:1) at room temperature for 30 min. Subsequently, the acrylamide solution was aspirated and gel polymerization solution (0.16% Ammonium persulfate and 0.2% TEMED in the 4% acrylamide solution) was added. Immediately, the tissues were covered with Gel Slick (Lonza #50640)-treated circular coverslips of 18 mm diameter (Ted Pella, 260369), transferred to an argon-filled chamber at room temperature and incubated for 30 min. After gel formation, the tissue sections were washed with 1x PBST twice and the coverslip was gently removed with a needle. At this point, the rolonies were crosslinked to the polyacrylamide gel.

## Image acquisition

Human brain DART-FISH sample was stained with 1x TrueBlack in 70% ethanol at room temperature for 2 min to reduce the lipofuscin auto-fluorescence and washed with 1x PBS three times for 3 min each before imaging. For the anchor round imaging, a mixture of anchor round probes, including DARTFISH_anchor_Cy3, dcProbe10_ATTO647N, and dcProbe7_AF488 probes, were diluted to 500 nM in 2x SSC and 30% formamide. Then, the samples were stained with anchor round probes at room temperature for 5 min and washed with 1 mL washing buffer (2x SSC, 10% formamide and 0.1% Tween-20) twice for 2 min each prior to imaging. The samples were immersed in 1 mL imaging buffer (2x SSC and 10% formamide) during imaging. For decoding imaging, each imaging cycle started with incubating samples with stripping buffer (2x SSC, 80% formamide, and 0.1% Tween-20) at room temperature for 5 min, washed with washing buffer twice for 2 min each, stained with a mixture of AlexaFluor488, Cy3, and ATTO647 fluorophore-labeled decoding probes (dcProbe0-AF488, dcProbe0-Cy3, and dcProbe0-ATTO647N as an example for round 1) in 2x SSC and 30% formamide for 10 min, and washed with washing buffer three times for 2 min each. Then, the samples were immersed in 1 mL of imaging buffer while imaging. After the last cycle of decoding imaging, DRAQ5 staining (5 µM, room temperature, 10 min) was performed for nuclei segmentation. Z-stack images were acquired by a resonant-scanning Leica TCS SP8 confocal microscope with 20x oil-immersion objective (NA = 0.75), pinhole size of 1 airy unit, pixel size of 284 nm x 284 nm (zoom=2) with 1024 ×1024 pixels per image, and 2 line averaging with 26 z-stacks (step size 1µm).

## DART-FISH image preprocessing and spot calling

The DART-FISH datasets were processed by our custom pipeline. The source codes of the pipeline can be found on GitHub (https://github.com/ypauling/human_brain_atlas_cortex_regionality) Raw z-stack images with 4 channels (3 fluorescent channels and brightfield) from the microscope were maximally-projected to the same z plane and registered to a reference round by affine transformation implemented in SimpleElastix[51] using the brightfield channel as the anchor. After the images were registered, the intensities were normalized by the maximum value in each image. Each field of view (FOV) underwent pixel-based decoding to identify the genes of each spot based on the barcodes assigned to the genes. Invalid barcodes were discarded. To obtain the global position of the rolonies, the FOVs were stitched by applying FIJI's[52] Grid/Collection Stitching plugin[53] (in headless mode) to the registered and maximum-projected brightfield images. Note that the theoretical positions of the FOVs, defined by the microscope, were used as initial positions for stitching. Nuclear boundaries were segmented with Cellpose (v2.1.1)[54,55]. The "nuc" model in Cellpose was used on DRAQ5 images (nuclei channel) to identify nuclear boundaries. Spots were assigned to the proximal nuclei within 15 µm.

## Spatial cell annotation

DART-FISH cells were annotated using an ad hoc method. We ran Tangram in the "cells" mode to align DART-FISH cells with pre-annotated SNARE-seq2 nuclei including 24 cell subclasses[9]. This provided us the probabilistic mapping of SNARE-seq2 nuclei in space

based on the cosine similarity of the expression of those 484 genes in DART-FISH cells. To do a deterministic cell annotation, we normalized the Tangram probabilities of each cell subclass and transferred the cell subclass label, having the highest normalized Tangram probability, to the DART-FISH cells. We applied additional filtering by the count of the marker genes in each annotated DART-FISH cell. The marker genes for each subclass used in this filtering step were summarized in Table S5.

### DART-FISH spatial distribution

The gaussian_kde function from scipy.stats was used to perform a kernel density estimation on each subclass in each region estimating the probability density function. For Pearson correlations, the estimated density was established for 1000 discrete units from pial surface to white matter and correlations across regions were determined from these values.

### DART-FISH neighborhood analysis

To identify the nearest neighbors of each cell subclass, we randomly sampled 100 cells belonging to each subclass as query cells. For a cell subclass having less than 100 cells, we sampled all cells belonging to that subclass for analysis. Cells within an Euclidean distance of 200 μm were called as nearest neighbors of the query cell subclass. For Pearson correlation calculations all neighborhood values were normalized to the proportion of all called cells in the profiled region of interest to account for broad variations in cellular composition across cortical regions. These normalized neighbor values for each subclass were then utilized to calculate Pearson correlation across cortical regions.

### Identifying genes/peaks that correlate with RC/TS axis

We removed lowly expressed genes by using the function filter_by_expr from python library decoupler (version 1.6.0) with the following parameters: min_count = 10 and min_total_count = 15. Similarly, we removed peaks of low coverage by the same function with the following parameters: min_count = 5 and min_total_count = 15. For each remaining gene/peak, we calculated the Spearman correlation score between expression/coverage and the order indices of regions along the Rostral-Caudal and the Transmodal-Sensorimotor axis. For genes, we used the following statistical cutoffs: adjusted p-value (BH corrected) <0.01 and absolute value of correlation score > 0.7. For peaks, the cutoffs are: adjusted p-value (BH corrected) <0.05 and absolute value of correlation score > 0.5. We used less stringent cutoffs for peaks to account for the fact that ATAC input matrices are more sparse.

### SCENIC+ network construction

We used SCENIC+ (version 1.0a1)[35] to construct regulatory networks for the complete dataset and multiple subclasses. To construct the cistopic object required for SCENIC + , we followed the tutorial https://pycistopic.readthedocs.io/en/latest/tutorials.html. We used the function evaluate_models from pycisTopic (version 2.0a0) to determine the optimal number of topics in each dataset. We then used various topic binarization methods (otsu, top3k) and region-specific cCREs called based on cell types in each subclass to identify potential enhancer regions. We used outputs from pycisTopic and RNA data as inputs to the SCENIC+ framework. We used default values for all adjustable parameters in the configuration file for the SCENIC+ pipeline. The specificity scores for each subclass are generated by the regulon_specificity_scores function from the scenicplus library using gene based AUC scores.

### Reporting summary

Further information on research design is available in the Nature Portfolio Reporting Summary linked to this article.

## Data availability

Data generated via SNARE-Seq2 can be found on the NEMO archive via the following link: https://data.nemoarchive.org/biccn/lab/zhang_kun/multimodal/sncell/ Associated metadata can be found in data S1. Data generated via DART-FISH is uploaded to the BIL archive and is freely accessible at the following https://doi.org/10.35077/g.1179. Source data are provided with this paper.

## Code availability

Code used for this manuscript can be found at the following github repository[56]: https://github.com/ypauling/human_brain_atlas_cortex_regionality.

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

## Acknowledgements

We would like to thank the research brain donors and families who shared the precious brain materials used in these studies for their deep contributions to science. We would like to thank the University of Washington Biorepository and Integrated Neuropathology (BRaIN) Laboratory. We would like to thank the Snohomish, Pierce, and King County Medical Examiner offices for their collaboration and support of the Pacific Northwest Brain Donor Network. We would like to thank Lisa Keene, Aimee Schantz, Emily Ragaglia, and the staff of the BRaIN Lab for outstanding research coordination and technical support and Dr. William Romanow for his scientific and technical support. Figures 1A, 4A, 5A, and S11C were created with the help of Biorender.com. The work was supported by the following grants: National Institutes of Health grant U01MH114828-01A1 (K.Z., P.K., B.R., J.C.), National Institutes of Health grant R01 AG065541 (J.C.), National Institutes of Health grant R01 AG071465 (J.C.), National Institutes of Health grant P30 AG066509 (C.D.K.), National Institutes of Health grant U19 AG072458 (C.D.K.), National Institutes of Health grant U19 AG060909 (C.D.K.), Allen Institute of Brain Sciences (C.D.K.), Nancy and Buster Alvord Endowment (C.D.K.).

## Author contributions

Conceptualization: C.P., J.S., B.Y., C.J.C., N.Z., J.C., P.K., B.R., K.Z. Methodology: C.P., J.S., C.J.C., K.C., N.P., D.D., Q.H., Software: J.S., B.Y., C.J.C., Q.H., Formal Analysis: C.P., J.S., B.Y., C.J.C., Data Curation: J.S., B.Y., C.J.C., Investigation: C.P., J.S., C.J.C., K.C., N.P., D.D., H.I., A.H., C.S.L., J.K., Q.H., R.H. Visualization: C.P., J.S., B.Y., C.J.C., N.Z., L.R., A.S., Funding acquisition: K.Z., J.C., P.K., B.R., Project administration: C.P., J.S., N.Z., C.S.L., R.D.H., C.D.K., E.L., J.C., B.R., K.Z., Supervision: N.Z., P.K.,

J.C., B.R., K.Z. Writing – original draft: C.P., Writing – review & editing: C.P., J.S., B.Y., C.J.C., N.Z., C.S.L., L.R., A.S., P.K., J.C., B.R., K.Z.

## Competing interests

B.R. is a co-founder of Epigenome Technologies, and has equity in Arima Genomics. J.C. has an employment relationship with Neurocrine Biosciences Inc., a company that may potentially benefit from the research results. Dr. Chun's relationship with Neurocrine Biosciences, Inc. has been reviewed and approved by Sanford Burnham Prebys Medical Discovery Institute in accordance with its Conflict of Interest Policies. The remaining authors declare no competing interests.
