## [Transparent Peer Review file · Nature Communications]

Single-cell multiomic human brain atlas reveals regulatory drivers of cortical regionality

Corresponding Author: Dr Kun Zhang

Version 0:

Reviewer comments:

Reviewer #1

(Remarks to the Author)

Thank you for the opportunity to review "Single-cell multiomic human brain atlas reveals regulatory drivers of cortical regionality" by Palmer and colleagues. The authors present a dataset of gene regulatory networks and transcriptional factors in nine regions. The main result is that several features, such as expression and chromatin accessibility, determine the differentiation between brain regions.

The study is ambitious and sure to be of wide interest to the field. The dataset and analysis is state of the art, and the question is of fundamental importance. I just have one question/suggestion, which the authors should consider optional:

1. The authors repeatedly invoke the term "gradient", which is in keeping with the broader molecular/cellular, literature, but I'd argue that it is difficult to claim that there is any sort of visible gradient in the present dataset given the very sparse spatial sampling of the cortex into 9 regions. If the authors feel that it is absolutely critical to refer to the spatial pattern as a gradient then that's fine, but I think that there is a more cautious road open, whereby they can speculate that their spatial pattern resembles the well-studied rostro-caudal and unimodal-transmodal gradients.

2. On a related note: it might be interesting to know what these 448-gene signatures actually look like on the whole brain. Would it be possible to take the values generated by DART-FISH and then plot them on the Allen Human Brain Atlas? This might actually help solidify the idea that the identified patterns follow a gradient.

Reviewer #2

(Remarks to the Author)

In this manuscript, Palmer and colleagues utilized joint single-cell RNA and ATAC profiling of dissected areas of the adult human cortex to identify region-specific signatures of cortical arealization. The findings from single-cell genomics experiments were then validated and extended using spatial transcriptomics. This work is an impressive technical feat and a resource that is of great value to the field. However, the paper can be improved, below are my comments:

1) Pseudobulk approaches used by authors have been shown to be underpowered (<https://www.nature.com/articles/s41467-021-21038-1>). Therefore, it probably misses a lot of important information. In addition to pseudobulk approach using edgeR and average gene expression profiles, it would be important to show how many of the DEGs are also detected using a single-cell approach (e.g. MAST) using a linear model including age, sex, PMI and RNA integrity, as well as donor information as a random effect model. If there is a good overlap between the two approaches, and the number of DEGs is higher in MAST analysis, these additional DEGs should at least be reported in a supplementary table.

2) Supplementary table 1 lists age and sex of donors but no additional information (e.g. postmortem interval, RNA integrity, manner of death). This information is important for anyone who uses the data generated by the authors.

3) The number of region-specific DEGs can be driven by the number of genes and read per cell across cell types (e.g. excitatory neurons are expected to have much higher number of genes expressed compared to say microglia). Authors need to normalize to reads/genes per cell in addition to down sampling cells. This goes for any analyses comparing number of DEGs or differential peaks between cell types.

4) "120 cell types" are not ever used in the paper after its first mention. I don't think it is informative to mention this very fine-grained classification since there are no analyses based on it.

5) Rostral-caudal gradients have been reported in the developing human cortex (e.g.

<https://www.nature.com/articles/s41586-021-03910-8>). It would be interesting and important to see whether any of the areal signatures observed by the authors have a developmental origin.

Version 1:

Reviewer comments:

Reviewer #1

(Remarks to the Author)

The authors have comprehensively addressed my concerns and I recommend publication.

Reviewer #2

(Remarks to the Author)

I thank the authors for their efforts in addressing the comments from the first review. All issues were addressed in full. No further comments from me.

We would like to thank the editor and both reviewers for their insightful and detailed comments. We have addressed the reviewers remarks below. Our responses to their comments are in BLUE below after each individual comment. Changes to the main and supplemental text are made in RED in the documents attached to our response. All following text in black are the entirety of the reviewers comments without edit.

=====

REVIEWER COMMENTS

Reviewer #1 (Remarks to the Author):

Thank you for the opportunity to review “Single-cell multiomic human brain atlas reveals regulatory drivers of cortical regionality” by Palmer and colleagues. The authors present a dataset of gene regulatory networks and transcriptional factors in nine regions. The main result is that several features, such as expression and chromatin accessibility, determine the differentiation between brain regions.

The study is ambitious and sure to be of wide interest to the field. The dataset and analysis is state of the art, and the question is of fundamental importance. I just have one question/suggestion, which the authors should consider optional:

1. The authors repeatedly invoke the term “gradient”, which is in keeping with the broader molecular/cellular, literature, but I’d argue that it is difficult to claim that there is any sort of visible gradient in the present dataset given the very sparse spatial sampling of the cortex into 9 regions. If the authors feel that it is absolutely critical to refer to the spatial pattern as a gradient then that’s fine, but I think that there is a more cautious road open, whereby they can speculate that their spatial pattern resembles the well-studied rostro-caudal and unimodal-transmodal gradients.

We thank the reviewer for this thoughtful suggestion. We agree that our dataset, based on nine cortical regions, provides an incomplete sampling of the cortical sheet and thus does not fully capture continuous spatial variation. We have revised the text to more cautiously describe these patterns, now referring to them as “*spatial patterns*”. We believe this framing better reflects the resolution of our dataset while still connecting to established cortical organizational principles.

2. On a related note: it might be interesting to know what these 448-gene signatures actually look like on the whole brain. Would it be possible to take the values generated by DART-FISH and then plot them on the Allen Human Brain Atlas? This might actually help solidify the idea that the identified patterns follow a gradient.

We appreciate this suggestion. While direct integration of our DART-FISH data with the Allen Human Brain Atlas would indeed be informative, our current dataset does not have the necessary feature alignments that permits a reliable projection, making a full-scale projection impossible, but we agree that spatial context is valuable. To address this, we have provided access to all imaging data (<https://doi.org/10.35077/g.1179>), allowing for detailed interrogation if desired. We also added text to highlight in the Discussion (pg 16) that future work integrating our dataset with whole-brain atlases such as Allen will be a valuable direction. Finally, we checked if any of the 448 genes used as DART-FISH probes were called as R-C correlated in the snRNA-seq and none of them were.

Reviewer #2 (Remarks to the Author):

In this manuscript, Palmer and colleagues utilized joint single-cell RNA and ATAC profiling of dissected areas of the adult human cortex to identify region-specific signatures of cortical arealization. The findings from single-cell genomics experiments were then validated and extended using spatial transcriptomics. This work is an impressive technical feat and a resource that is of great value to the field. However, the paper can be improved, below are my comments:

1) Pseudobulk approaches used by authors have been shown to be underpowered (<https://www.nature.com/articles/s41467-021-21038-1>). Therefore, it probably misses a lot of important information. In addition to pseudobulk approach using edgeR and average gene expression profiles, it would be important to show how many of the DEGs are also detected using a single-cell approach (e.g. MAST) using a linear model including age, sex, PMI and RNA integrity, as well as donor information as a random effect model. If there is a good overlap between the two approaches, and the number of DEGs is higher in MAST analysis, these additional DEGs should at least be reported in a supplementary table.

We thank the reviewer for raising this important point. In addition to our pseudobulk differential expression analyses, we have now performed complementary single-cell level testing using MAST and have incorporated this analysis into the text (pg 4). Details on the new bioinformatic analysis have also been added to the methods section (pg 18).

We found substantial overlap between pseudobulk and single-cell analyses (69.9% of all region specific DEGs from EdgeR were also found in MAST). The MAST analysis also revealed additional DEGs, which we now provide in Supplementary Table 3. This analysis confirms that our findings are robust to both approaches.

Added text on pg 4 is "To account for the possibility that pseudobulk approaches can be underpowered we utilized MAST to account for age, sex, PMI, and RIN effects on differentially expressed genes (Supplementary Data 3). As expected, MAST identified more regional DEGs, 21,234 as compared to 4,101 with edgeR, but 69.9% of the edgeR genes were also called as regional DEGs by MAST, confirming the robust nature of the analysis."

2) Supplementary table 1 lists age and sex of donors but no additional information (e.g. postmortem interval, RNA integrity, manner of death). This information is important for anyone who uses the data generated by the authors.

We agree that extended donor metadata is important for reuse of the data presented in the manuscript and appreciate the reviewer catching this oversight. We have now added postmortem interval, RNA integrity number, and manner of death to Supplementary Table 1.

3) The number of region-specific DEGs can be driven by the number of genes and read per cell across cell types (e.g. excitatory neurons are expected to have much higher number of genes expressed compared to say microglia). Authors need to normalize to reads/genes per cell in addition to down sampling cells. This goes for any analyses comparing number of DEGs or differential peaks between cell types.

We thank the reviewer for highlighting this point. To address this in a robust manner, we have normalized edgeR subclass DEG counts to the total genes that pass the filter for use in edgeR

analysis in that given subclass on a downsampled subset of cells. Stated differently, for a constant number of cells from each subclass and region we identified the total number of genes that pass the edgeR filter for analysis (edgeR_subclass_total) and the called edgeR DEGs (edgeR_Total_DEGs) and calculated a normalized value as (edgeR_Total_DEGs)/(edgeR_subclass_total). We have plotted these values and added them to (Supplementary Fig. 4b) in addition to adding text (pg 4) and methods clarification. The two plots have nearly identical trends supporting the fact that the number of DEGs is robustly varied by cell subclass.

We believe that this is the most robust quantification of the total number of genes for a given subclass. Our key conclusions remain unchanged: excitatory neurons show the strongest areal signatures.

We completed a similar analysis for region-specific cCREs and again conclusions remained intact and we have incorporated this into the text (pg 6) as well as a new Supplementary Fig. 6a.

4) “120 cell types” are not ever used in the paper after its first mention. I don’t think it is informative to mention this very fine-grained classification since there are no analyses based on it.

We thank the reviewer for this thoughtful comment. While we agree that we do not perform extensive downstream analyses at the cell-type level outlined in Figure 1e, we believe that including this classification is valuable for several reasons. First, as the reviewer noted, our study is intended to serve as a resource for the community. By classifying cells at the type level using standardized nomenclature, we provide a framework that will facilitate future analyses and save time for other researchers working with these data. Second, it is important to demonstrate the reproducibility of standardized taxonomies. Our classifications show strong concordance with previously published cortical taxonomies (Jorstad 2023), reinforcing the robustness of our dataset. Finally, to further highlight the relevance of this classification, we have expanded the Results section (pg 4) to describe the biological insights revealed by Figure 1e. New text is as follows “Clustering identified 120 cell types (Fig. 1e) in accordance with previous efforts and offered insight into V1C enriched glutamatergic cell types as well as the significant cell-type diversity contained within SST and VIP subclasses.”

5) Rostral-caudal gradients have been reported in the developing human cortex (e.g. <https://www.nature.com/articles/s41586-021-03910-8>). It would be interesting and important to see whether any of the areal signatures observed by the authors have a developmental origin.

This is an excellent suggestion from the reviewer and we appreciate it. We examined the overlap between our adult cortical areal signatures and developmental arealization programs described by Bhaduri et al. (2021, Nature). To compare the datasets, we interrogated the region enriched genes as defined in supplementary table 8 from the referenced manuscript to our data. We only interrogated genes called as region specific in neurons (as radial glial cells do not have a direct comparison in our dataset). Any gene identified as region specific in neurons in any brain from the referenced work was checked if it was also regionally enriched in any neuronal cell subclass in our datasets. The overlap was minimal, but present. We found that ~5-15% of adult areal DEGs overlap with genes differentially expressed during development. This overlap is shown in Supplementary Fig. 4e. Notably there are key transcription factors, including *NFIA*, *NFIX*, *TCF4*, and *NR2F1*, that are conserved, further highlighting their essential roles in

neuronal spatial specialization. These results suggest that a subset of adult cortical areal signatures may indeed reflect developmental patterning. We now discuss this point in the Results (pg 6) and Discussion (pg 16) and have included the overlapping genes in Supplementary Data 2.

We would like to thank the editor and both reviewers for their insightful and detailed comments. We have addressed the reviewers remarks below. Our responses to their comments are in **BLUE** below after each individual comment. Changes to the main and supplemental text are made in **RED** in the documents attached to our response. All following text in black are the entirety of the reviewers comments without edit.

REVIEWER COMMENTS

Reviewer #1 (Remarks to the Author):

Thank you for the opportunity to review “Single-cell multiomic human brain atlas reveals regulatory drivers of cortical regionality” by Palmer and colleagues. The authors present a dataset of gene regulatory networks and transcriptional factors in nine regions. The main result is that several features, such as expression and chromatin accessibility, determine the differentiation between brain regions.

The study is ambitious and sure to be of wide interest to the field. The dataset and analysis is state of the art, and the question is of fundamental importance. I just have one question/suggestion, which the authors should consider optional:

1. The authors repeatedly invoke the term “gradient”, which is in keeping with the broader molecular/cellular, literature, but I’d argue that it is difficult to claim that there is any sort of visible gradient in the present dataset given the very sparse spatial sampling of the cortex into 9 regions. If the authors feel that it is absolutely critical to refer to the spatial pattern as a gradient then that’s fine, but I think that there is a more cautious road open, whereby they can speculate that their spatial pattern resembles the well-studied rostral-caudal and unimodal-transmodal gradients.

We thank the reviewer for this thoughtful suggestion. We agree that our dataset, based on nine cortical regions, provides an incomplete sampling of the cortical sheet and thus does not fully capture continuous spatial variation. We have revised the text to more cautiously describe these patterns, now referring to them as “*spatial patterns*”. We believe this framing better reflects the resolution of our dataset while still connecting to established cortical organizational principles.

2. On a related note: it might be interesting to know what these 448-gene signatures actually look like on the whole brain. Would it be possible to take the values generated by DART-FISH and then plot them on the Allen Human Brain Atlas? This might actually help solidify the idea that the identified patterns follow a gradient.

We appreciate this suggestion. While direct integration of our DART-FISH data with the Allen Human Brain Atlas would indeed be informative, our current dataset does not have the necessary feature alignments that permits a reliable projection, making a full-scale projection impossible, but we agree that spatial context is valuable. To address this, we have provided access to all imaging data (<https://doi.org/10.35077/g.1179>), allowing for detailed interrogation if desired. We also added text to highlight in the Discussion (pg 16) that future work integrating our dataset with whole-brain atlases such as Allen will be a valuable direction. Finally, we checked if any of the 448 genes used as DART-FISH probes were called as R-C correlated in the snRNA-seq and none of them were.

Reviewer #2 (Remarks to the Author):

In this manuscript, Palmer and colleagues utilized joint single-cell RNA and ATAC profiling of dissected areas of the adult human cortex to identify region-specific signatures of cortical arealization. The findings from single-cell genomics experiments were then validated and extended using spatial transcriptomics. This work is an impressive technical feat and a resource that is of great value to the field. However, the paper can be improved, below are my comments:

1) Pseudobulk approaches used by authors have been shown to be underpowered (<https://www.nature.com/articles/s41467-021-21038-1>). Therefore, it probably misses a lot of important information. In addition to pseudobulk approach using edgeR and average gene expression profiles, it would be important to show how many of the DEGs are also detected using a single-cell approach (e.g. MAST) using a linear model including age, sex, PMI and RNA integrity, as well as donor information as a random effect model. If there is a good overlap between the two approaches, and the number of DEGs is higher in MAST analysis, these additional DEGs should at least be reported in a supplementary table.

We thank the reviewer for raising this important point. In addition to our pseudobulk differential expression analyses, we have now performed complementary single-cell level testing using MAST and have incorporated this analysis into the text (pg 4). Details on the new bioinformatic analysis have also been added to the methods section (pg 18).

We found substantial overlap between pseudobulk and single-cell analyses (69.9% of all region specific DEGs from EdgeR were also found in MAST). The MAST analysis also revealed additional DEGs, which we now provide in Supplementary Table 3. This analysis confirms that our findings are robust to both approaches.

Added text on pg 4 is "To account for the possibility that pseudobulk approaches can be underpowered we utilized MAST to account for age, sex, PMI, and RIN effects on differentially expressed genes (Supplementary Data 3). As expected, MAST identified more regional DEGs, 21,234 as compared to 4,101 with edgeR, but 69.9% of the edgeR genes were also called as regional DEGs by MAST, confirming the robust nature of the analysis."

2) Supplementary table 1 lists age and sex of donors but no additional information (e.g. postmortem interval, RNA integrity, manner of death). This information is important for anyone who uses the data generated by the authors.

We agree that extended donor metadata is important for reuse of the data presented in the manuscript and appreciate the reviewer catching this oversight. We have now added postmortem interval, RNA integrity number, and manner of death to Supplementary Table 1.

3) The number of region-specific DEGs can be driven by the number of genes and read per cell across cell types (e.g. excitatory neurons are expected to have much higher number of genes expressed compared to say microglia). Authors need to normalize to reads/genes per cell in addition to down sampling cells. This goes for any analyses comparing number of DEGs or differential peaks between cell types.

We thank the reviewer for highlighting this point. To address this in a robust manner, we have normalized edgeR subclass DEG counts to the total genes that pass the filter for use in edgeR

analysis in that given subclass on a downsampled subset of cells. Stated differently, for a constant number of cells from each subclass and region we identified the total number of genes that pass the edgeR filter for analysis (edgeR_subclass_total) and the called edgeR DEGs (edgeR_Total_DEGs) and calculated a normalized value as (edgeR_Total_DEGs)/(edgeR_subclass_total). We have plotted these values and added them to (Supplementary Fig. 4b) in addition to adding text (pg 4) and methods clarification. The two plots have nearly identical trends supporting the fact that the number of DEGs is robustly varied by cell subclass.

We believe that this is the most robust quantification of the total number of genes for a given subclass. Our key conclusions remain unchanged: excitatory neurons show the strongest areal signatures.

We completed a similar analysis for region-specific cCREs and again conclusions remained intact and we have incorporated this into the text (pg 6) as well as a new Supplementary Fig. 6a.

4) “120 cell types” are not ever used in the paper after its first mention. I don’t think it is informative to mention this very fine-grained classification since there are no analyses based on it.

We thank the reviewer for this thoughtful comment. While we agree that we do not perform extensive downstream analyses at the cell-type level outlined in Figure 1e, we believe that including this classification is valuable for several reasons. First, as the reviewer noted, our study is intended to serve as a resource for the community. By classifying cells at the type level using standardized nomenclature, we provide a framework that will facilitate future analyses and save time for other researchers working with these data. Second, it is important to demonstrate the reproducibility of standardized taxonomies. Our classifications show strong concordance with previously published cortical taxonomies (Jorstad 2023), reinforcing the robustness of our dataset. Finally, to further highlight the relevance of this classification, we have expanded the Results section (pg 4) to describe the biological insights revealed by Figure 1e. New text is as follows “Clustering identified 120 cell types (Fig. 1e) in accordance with previous efforts and offered insight into V1C enriched glutamatergic cell types as well as the significant cell-type diversity contained within SST and VIP subclasses.”

5) Rostral-caudal gradients have been reported in the developing human cortex (e.g. <https://www.nature.com/articles/s41586-021-03910-8>). It would be interesting and important to see whether any of the areal signatures observed by the authors have a developmental origin.

This is an excellent suggestion from the reviewer and we appreciate it. We examined the overlap between our adult cortical areal signatures and developmental arealization programs described by Bhaduri et al. (2021, Nature). To compare the datasets, we interrogated the region enriched genes as defined in supplementary table 8 from the referenced manuscript to our data. We only interrogated genes called as region specific in neurons (as radial glial cells do not have a direct comparison in our dataset). Any gene identified as region specific in neurons in any brain from the referenced work was checked if it was also regionally enriched in any neuronal cell subclass in our datasets. The overlap was minimal, but present. We found that ~5-15% of adult areal DEGs overlap with genes differentially expressed during development. This overlap is shown in Supplementary Fig. 4e. Notably there are key transcription factors, including *NFIA*, *NFIX*, *TCF4*, and *NR2F1*, that are conserved, further highlighting their essential roles in

neuronal spatial specialization. These results suggest that a subset of adult cortical areal signatures may indeed reflect developmental patterning. We now discuss this point in the Results (pg 6) and Discussion (pg 16) and have included the overlapping genes in Supplementary Data 2.